

# (Re)shaping online narratives: when bots promote the message of President Trump during his first impeachment

Michael C. Galgoczy[1], Atharva Phatak[2], Danielle Vinson[3], Vijay K. Mago[2] and Philippe J. Giabbanelli[1]

[1] Department of Computer Science & Software Engineering, Miami University, Oxford, OH, United States
[2] Department of Computer Science, Lakehead University, Thunder Bay, ON, Canada
[3] Department of Politics & International Affairs, Furman University, Greenville, SC, United States

Corresponding author
Philippe J. Giabbanelli,
giabbapj@miamioh.edu

## ABSTRACT

Influencing and framing debates on Twitter provides power to shape public opinion. Bots have become essential tools of 'computational propaganda' on social media such as Twitter, often contributing to a large fraction of the tweets regarding political events such as elections. Although analyses have been conducted regarding the first impeachment of former president Donald Trump, they have been focused on either a manual examination of relatively few tweets to emphasize rhetoric, or the use of Natural Language Processing (NLP) of a much larger *corpus* with respect to common metrics such as sentiment. In this paper, we complement existing analyses by examining the role of bots in the first impeachment with respect to three questions as follows. (Q1) Are bots actively involved in the debate? (Q2) Do bots target one political affiliation more than another? (Q3) Which sources are used by bots to support their arguments? Our methods start with collecting over 13M tweets on six key dates, from October 6th 2019 to January 21st 2020. We used machine learning to evaluate the sentiment of the tweets (*via* BERT) and whether it originates from a bot. We then examined these sentiments with respect to a balanced sample of Democrats and Republicans directly relevant to the impeachment, such as House Speaker Nancy Pelosi, senator Mitch McConnell, and (then former Vice President) Joe Biden. The content of posts from bots was further analyzed with respect to the sources used (with bias ratings from AllSides and Ad Fontes) and themes. Our first finding is that bots have played a significant role in contributing to the overall negative tone of the debate (Q1). Bots were targeting Democrats more than Republicans (Q2), as evidenced both by a difference in ratio (bots had more negative-to-positive tweets on Democrats than Republicans) and in composition (use of derogatory nicknames). Finally, the sources provided by bots were almost twice as likely to be from the right than the left, with a noticeable use of hyper-partisan right and most extreme right sources (Q3). Bots were thus purposely used to promote a misleading version of events. Overall, this suggests an intentional use of bots as part of a strategy, thus providing further confirmation that computational propaganda is involved in defining political events in the United States. As any empirical analysis, our work has several limitations. For example, Trump's rhetoric on Twitter has previously been characterized by an overly negative tone, thus tweets detected as negative may be echoing his message rather than acting against him. Previous works show that this possibility is limited, and its existence would only strengthen our conclusions. As our analysis is based on NLP, we focus on processing a large volume

of tweets rather than manually reading all of them, thus future studies may complement our approach by using qualitative methods to assess the specific arguments used by bots.

## INTRODUCTION

The efforts of party leaders and presidents to shape public understanding of issues through mainstream and partisan media are well documented (*Sellers, 2009*; *Vinson, 2017*). Partisan media intentionally adopt ideological frames and cover stories in ways that favor the politicians in their own party and create negative impressions of those in the other party, contributing to polarization in American politics (*Levendusky, 2013*; *Forgette, 2018*). At the interface of political communication and computational sciences, the emerging field of computational politics has produced many analyses of polarization over the recent years (*Haq et al., 2020*; *Pozen, Talley & Nyarko, 2019*; *Yarchi, Baden & Kligler-Vilenchik, 2021*). Comparatively less computational research has been devoted to how politicians influence debate on social media, but interest in this question has grown during Donald Trump's presidency as he extensively communicated directly to the public *via* Twitter (*Ouyang & Waterman, 2020*).

While mainstream media traditionally favor those with political power, giving them coverage that expands their public reach, social media rewards those who can attract an audience. On Twitter, a large attentive audience can amplify a politician's message and change or mobilize public opinion (*Zhang et al., 2018*; *Russell, 2021*). Trump's rhetoric features many characteristics of populism, including emotional appeals based on opinion rather than fact and allegations of corruption, that attracts followers; indeed, his messages are often picked up by right-wing populist leaders around the world (*Pérez-Curiel, Rivas-de Roca & García-Gordillo, 2021*). As research on the 2016 presidential campaign illustrates, Trump had an attentive audience, particularly on the far right, that readily amplified his message through retweets (*Zhang et al., 2018*); a similar phenomenon was found in the 2020 election, for which datasets are gradually assembled and examined (*Abilov et al., 2021*; *Grimminger & Klinger, 2021*). These events repeatedly show that news media responded to Trump's large volume of tweets by giving him substantially more coverage than other candidates, further expanding his reach to influence and frame public debate (*Wells et al., 2020*). Computational analyses further suggest that the use of tweets was a strategic activity, as Trump's Twitter activity increased in response to waning news coverage from Center Right, Mainstream, Left Wing, and Far Left media (*Wells et al., 2020*). Influencing and framing the debate on Twitter thus serves to shape media coverage, public opinion, and ultimately impact the political agenda (*Şahin, Johnson & Korkut, 2021*).

In this sense, shaping public discourse is connected to political power. Studies of Twitter rhetoric indicate that patterns in framing and use of language occur within digital networks that create group identities (*Alfonzo, 2021*). Critical discourse analysis suggests that examining messaging can identify power asymmetries, revealing the institutions and leaders who control public discourse in particular environments or around specific issues and events (*Van Dijk, 2009*). Such research can reveal abuse of power by dominant groups and resistance by marginalized groups. Given Trump's populist rhetoric and its potential for spreading disinformation (*Pérez-Curiel, Rivas-de Roca & García-Gordillo, 2021*), amplification of his messages would have important implications for public discourse. Though critical analysis would require a more detailed and nuanced coding of the tweets than we are able to provide with our methodology, looking at the amplification of political discourse by bots allows us to see the potential manipulation of public discourse, especially if the bots favor one faction, that could affect the power dynamics between parties and have significant negative implications for democracy. This kind of power merits additional study. Our study of the first impeachment of Donald Trump provides such an opportunity.

This paper is part of the growing empirical literature on the social influence carried by Trump's tweets prior to the permanent suspension of his account on 8 January 2021. Such studies are now covered by dedicated volumes (*Kamps, 2021*; *Schier, 2022*) as well as a wealth of scholarly works, including large-scale social network analysis and mining (*Zheng et al., 2021*), analyses of media responses (*Christenson, Kreps & Kriner, 2021*), the interplay of tweets and media (*Morales, Schultz & Landreville, 2021*), or the (absence of) impact between his tweets and stock returns (*Machus, Mestel & Theissen, 2022*). Related studies also include analyses about comments from political figures *about Trump* (*Milford, 2021*; *Alexandre, Jai-sung Yoo & Murthy, 2021*) and *vice versa* (*Brown Crosby, 2022*), analyses of his letter during the first impeachment (*Reyes & Ross, 2021*), and detailed examinations about how Trump's message was perceived by specific groups such as White extremists (*Long, 2022*). The impeachment and handling of COVID-19 have received particular attention on Twitter (*Dejard et al., 2021*; *Cervi, García & Marín-Lladó, 2021*), as key moments of his presidency. While the use of bots to create social media content during presidential elections and COVID-19 have received attention (*Xu & Sasahara, 2021*), there is still a need to examine how bots were used during the first impeachment.

The specific contribution of our study is to complement existing (predominantly manual) analyses of tweets during the first impeachment, which often focused on tweets from Donald Trump (*Gould, 2021*; *Driver, 2021*) or Senators (*McKee, Evans & Clark, 2022*), by using machine learning to examine the role of bots in a large collection of tweets. This complementary analysis has explicitly been called for in recent publications (*Tachaiya et al., 2021*), which performed large-scale sentiment analyses during the first impeachment (*via* Reddit and 4chan), but did not investigate the efforts at social

engineering deployed *via* bots. Our specific contribution focuses on addressing three specific questions:

(Q1) Are bots actively involved in the debate?
(Q2) Do bots target one political affiliation more than another?
(Q3) Which sources are used by bots to support their arguments?

The remainder of this paper is organized as follows. In the next section, we explain how we gathered over 13M tweets based on six key dates of the first impeachment, and analyzed them using BERT (for sentiment) and Botometer (for bot detection) as well as bias ratings (for source evaluation) and Latent Dirichlet Allocation (for theme mining). We then present our results for a balanced sample of key political figures in the impeachment (including Adam Schiff, Joe Biden, Mitch McConnell, Nancy Pelosi, Donald Trump) as well as others relevant to our analysis. For transparency and replicability of research, tweets and detailed results are available on a third-party repository without registration needed, at https://osf.io/3nsf8/. Finally, our discussion contextualizes the answer for each of the three questions within the broader literature on computational politics and summarizes some of our limitations.

## METHODS

### Data collection

For a detailed description of the Ukraine Whistleblower Scandal, we refer the readers to a recent analysis by McKee and colleagues who also contextualize this situation within the broader matter of hyper-polarization (*McKee, Evans & Clark, 2022*). A short overview is as follows. On 9 September 2019, the House Intelligence Committee, headed by Rep. Adam Schiff, was alerted to the existence of a whistleblower complaint that involved the Trump Administration. Over the next 2 weeks, information from media reports indicated that President Trump had withheld aid to Ukraine possibly to pressure the new Ukrainian President to launch a corruption investigation into Democratic presidential hopeful Joe Biden's son Hunter who had been on the board of a Ukrainian company when the elder Biden was Vice President during the Obama Administration (*Trautman, 2019*). By 25th September, House Speaker Nancy Pelosi, a Democrat, had launched an impeachment inquiry, and the Trump Administration had released notes from Trump's call with the Ukrainian President that some saw as evidence that Trump had conditioned aid and a visit to the White House on Ukraine investigating Hunter Biden. Our data collection started soon after, covering six key dates leading to the impeachment process.

On 6 October 2019, a second whistleblower was identified and we collected 1,968,943 tweets. On 17 October 2019, then Senate Majority Leader Mitch McConnell briefed the Senate on the impeachment process (1,960,808 tweets). On 14 November 2019, the second day of impeachment hearings began in the House Intelligence Committee (1,977,712 tweets). On 5 December 2019, 1 day after the House Judiciary Committee opened impeachment hearings, four constitutional lawyers testified in front of Congress (1,960,813 tweets). On 18 December 2019, the House debated and passed two articles of

**Table 1 Distribution of tweets collected.**

| Date | Tweets collected |
|---|---|
| 6 October 2019 | 1,968,943 |
| 17 October 2019 | 1,960,808 |
| 14 November 2019 | 1,977,712 |
| 5 December 2019 | 1,960,813 |
| 19 December 2019 | 2,041,924 |
| 21 January 2020 | 3,360,434 |

impeachment. We collected 2,041,924 to gather responses on the following day. Finally, on 21 January 2020, the Senate impeachment trial began and we ended the data collection with 3,360,434 tweets. In total, 13,272,577 tweets were collected (Table 1).

Throughout the process, both parties and the President tried to control the public narrative over impeachment. Early in the process, Democrats began to talk about abuse of power or corruption by Trump and eventually shifted to the more specific language of bribery and extortion. Meanwhile, Republican elected officials seemed to be trying to frame this as a coup by the Democrats. Republicans directed their ire at House Intelligence Committee Chairman Democrat Adam Schiff who led the impeachment inquiry. The President and some of his Republican allies, including Senator Lindsey Graham and aide Stephen Miller, tried to divert attention from the President by alleging misconduct by Joe and Hunter Biden. Our key words for data collection were thus chosen to reflect the messaging of the two parties in what was shaping up as a largely partisan process: Trump, Impeachment, Coup, Abuse of Power, Schiff (as in Adam), and Biden. We did not explicitly include words like disinformation, though the messaging may have included unfounded allegations, because such words were not part of either party's framing.

Tweets were gathered using our Twitter mining system (Mendhe et al., 2020), which allows for the collection of social media data and the application of many filters for cleaning and further use for machine learning techniques.

## Preprocessing

This study and the methods employed only target texts in English, hence we removed tweets in other languages as well as tweets only containing hyperlinks. Language detection of Twitter data is more difficult than that of standard texts due to the unusual language used within tweets. We used the `langdetect` library (Table 2), which utilizes a naive Bayes classifier and achieves strong performances on tweets (Lui & Baldwin, 2014). Tweets in English may have to undergo extensive cleaning through a variety of filters, depending on the model used for analysis. As detailed in the next section, we use the BERT model, or **B**idirectional **E**ncoder **R**epresentations from **T**ransformers. BERT is pre-trained on *unprocessed* English text (Devlin et al., 2018) hence the only processing performed was to remove tweets that consisted of only hyperlinks. A more detailed discussion of our data collection platform and its combination with BERT can be found in (Qudar & Mago, 2020),

**Table 2 Language detection using `langdetect`.**

| Tweet | Output |
|---|---|
| tüm dış politikayı tek bir adama yani trump'a bağlamak tüm | tr |
| I'm sure Trump will tweet about rep Elijah Cummings passing | en |

**Table 3 Nicknames of key actors.**

| Actor | Nicknames |
|---|---|
| Nancy Pelosi | Nervous Nancy |
| Adam Schiff | Shifty Schiff |
| Joe Biden | Sleepy Joe, Creepy Joe |
| Mitch McConnell | Midnight Mitch |

while an abundance of studies show how BERT is used for classification of tweets (*Singh, Jakhar & Pandey, 2021*; *Yenduri et al., 2021*; *Sadia & Basak, 2021*).

## Analysis

In textual data analysis, we can apply information extraction and text mining to parse information from a large data set. In order to understand how each side put their messages out, data was broken down by mention of specific politicians–those who were the subject of impeachment (Trump) and countermessaging by Republicans (Biden), those who were important to the impeachment process and messaging for their party (Pelosi, McConnel, and Schiff), and two Republicans (Miller and Graham) "who represent accounts closely associated with President Trump" per previous Twitter analyses (*Boucher & Thies, 2019*; *Hawkins & Kaltwasser, 2018*). This sample of actors is evenly divided among Democrats and Republicans. After assigning tweets to each of the actors, we then used sentiment analysis and bot detection. The analysis was further nuanced by accounting for the use of a derogatory nickname (Table 3) in lieu of the official actor's name. In addition, we performed complementary examinations of the content of tweets by bots, by examining their themes (in contrast with human tweets) and the bias exhibited in the websites used as sources.

In summary, the data associated with each tweet (content, time, author) is automatically expanded with the list of actors involved, the type of author (bot, human, unknown), the sentiment (negative, neutral, positive), and the list of websites used.

### Sentiment analysis

Sentiment analysis is a very important technique to analyze textual data and is frequently used in natural language processing, text analysis, and computational linguistics. Our model classifies sentiment into three categories: negative, neutral, and positive. As mentioned above, we use the natural language processing model BERT, designed to pre-train deep bidirectional representations from unlabeled text by jointly conditioning on both left and right context in all layers (*Devlin et al., 2018*). As its name indicates, BERT is a

pre-trained deep bidirectional transformer, whose architecture consists of multiple encoders, each composed of two types of layers (multi-head self-attention layers, feed forward layers). To appreciate the number of parameters, consider that the text first goes through an embedding process (two to three dozen million parameters depending on the model), followed by transformers (each of which adds 7 or 12.5 million parameters depending on the model), ending with a pooling layer (0.5 or 1 million more parameters depending on the model). All of these parameters are trainable. Although the model has already been pre-trained, it is common practice to give it additional training based on data specific to the context of interest.

In order to train the model, a set of 3,000 unprocessed tweets was randomly selected, distributed evenly across each of the six collection dates. The tweets were then manually labelled for overall sentiment, and fed into the model for training. This approach is similar to other works, such as *Grimminger & Klinger (2021)*, in which 3,000 tweets were manually annotated for a BERT classification. While there is no universal number for how many tweets should be labeled when using BERT, we observe a range across studies from under 2,000 labeled tweets (*Şaşmaz & Tek, 2021*; *Alomari, ElSherif & Shaalan, 2017*; *Peisenieks & Skadiņš, 2014*) up to several thousands (*Golubev & Loukachevitch, 2020*; *Rustam et al., 2021*; *Nabil, Aly & Atiya, 2015*; *Zhang et al., 2020*). In line with previous studies on sentiment analysis in tweets (*Rustam et al., 2021*; *Khan et al., 2021*), we compared BERT and several other supervised machine learning models from the `textblob` library (*i.e.*, decision tree, Naive Bayes) as well as scikit-learn (logistic regression, support vector machine). Our comparison in Table 4 confirms that BERT leads to satisfactory and more stable performances than alternatives, thus validating the choice of BERT in our corpus.

### Bot detection

We use the `botometer` library (formerly know as `BotOrNot`) to detect the presence of bots in Twitter. `botometer` is a paid service that utilizes the Twitter API to compile over 1,000 features of a given Twitter account, including how frequently the account tweets, how similar each tweet is to previous tweets from the account, and the account's ratio of followers to followees (*Davis et al., 2016*). Features are examined both within their respective categories and collectively. The resulting series of probabilities is returned (Table 5). To provide a conservative estimate, we scanned almost half of the different accounts associated with tweets (1,101,023 out of 2,438,343 unique accounts). Since some accounts were deleted or suspended ($n = 113,037$), our analysis of bots is based on a sample of 987,987 active accounts. We note that the percentage of accounts unavailable is similar to *Le et al. (2019)*, in which 9.5% of Twitter accounts involved in the 2016 election were suspended.

In order to classify Twitter accounts as human or bot, a classifier needs to be properly calibrated on annotated botometer data (*Spagnuolo, 2019*). We considered three supervised machine learning techniques, which are frequently used alongside BERT to process tweets: decision trees, extreme gradient boosted trees (XGBoost), and random forests (*Kumar et al., 2021*; *Rustam et al., 2021*). All three methods were implemented in

**Table 4 Comparison of classification accuracy for BERT (which we use in this study), TextBlob library, and two scikit-learn algorithms.**

| Model name | Class | Metric | Value |
|---|---|---|---|
| Decision Tree | Overall | Accuracy | 0.62 |
| | | Precision | 0.64 |
| | | Recall | 0.8816 |
| | Negative | F1 | 0.744 |
| | | Precision | 0.537 |
| | | Recall | 0.2562 |
| | Neutral | F1 | 0.3465 |
| | | Precision | 0.478 |
| | | Recall | 0.2104 |
| | Positive | F1 | 0.2922 |
| Naive Bayes | Overall | Accuracy | 0.658 |
| | | Precision | 0.664 |
| | | Recall | 0.918 |
| | Negative | F1 | 0.7707 |
| | | Precision | 0.638 |
| | | Recall | 0.3607 |
| | Neutral | F1 | 0.4608 |
| | | Precision | 0.617 |
| | | Recall | 0.0979 |
| | Positive | F1 | 0.1686 |
| BERT | Overall | Accuracy | 0.7398 |
| | | Precision | 0.821 |
| | | Recall | 0.806 |
| | Negative | F1 | 0.813 |
| | | Precision | 0.6246 |
| | | Recall | 0.666 |
| | Neutral | F1 | 0.6348 |
| | | Precision | 0.612 |
| | | Recall | 0.584 |
| | Positive | F1 | 0.598 |
| Logistic Regressor | Overall | Accuracy | 0.635 |
| | | Precision | 0.652 |
| | | Recall | 0.9 |
| | Negative | F1 | 0.7535 |
| | | Precision | 0.543 |
| | | Recall | 0.303 |
| | Neutral | F1 | 0.389 |
| | | Precision | 0.72 |
| | | Recall | 0.13125 |
| | Positive | F1 | 0.217 |

| Table 4 (continued) | | | |
| --- | --- | --- | --- |
| Model name | Class | Metric | Value |
| Support Vector Machine | Overall | Accuracy | 0.64 |
| | | Precision | 0.639 |
| | | Recall | 0.94 |
| | Negative | F1 | 0.762 |
| | | Precision | 0.637 |
| | | Recall | 0.238 |
| | Neutral | F1 | 0.3465 |
| | | Precision | 0.665 |
| | | Recall | 0.128 |
| | Positive | F1 | 0.199 |

Table 5 Botometer results.

| Twitter account ID | English | Content | … | Universal |
| --- | --- | --- | --- | --- |
| 999800336750530560 | 0.067063 | 0.938607 | … | 0.536176 |
| 306127388 | 0.046774 | 0.833483 | … | 0.207906 |

scikit-learn, and our scripts are provided publicly on our repository. Training data, consisting of manually labelled Twitter accounts, was retrieved from https://botometer. osome.iu.edu/bot-repository/datasets.html and compiled, thus providing 3,000 accounts. For each of the three supervised machine learning training, we optimized a classifier using 10-fold cross validation and a grid search over commonly used hyper-parameters. The hyper-parameters considered and their values are listed in Table 6, while each model was exported from scikit-learn and saved on our open repository for inspection and reuse by the research community. As performances are comparable across the three techniques, we used the decision tree (which can more readily be interpreted by experts) to classify accounts from our collected data.

### Content of tweets by Bots

Once we detect the posts emanating from bots, we can further examine their content. In particular, we use a frequency counts approach to measure the number of times that each website is cited in the bots' tweets. Specifically, we retrieve all URLs from these tweets and map them to a common form such that 'www.foxnews.com', 'media2.foxnews.com' or 'radio.foxnews.com' are all counted as foxnews. Then, we examine the political leanings of the websites by cross-referencing them with sources for bias ratings. We accomplish this in the same manner as in the recent work of *Huszár et al. (2022)*, which used two sources for bias ratings: AllSides (https://www.allsides.com/media-bias/ratings) and Ad Fontes (https://adfontesmedia.com/interactive-media-bias-chart). In line with their work, we do not claim that either source provides objectively better ratings (*Huszár et al., 2022*), hence we report both. If a website is rated by neither source, then we access it to read its content and evaluate it manually; such evaluations are disclosed explicitly.

**Table 6 Classification performances on bot detection.**

| Approach | Hyper-parameters | | Performances | | | | |
| --- | --- | --- | --- | --- | --- | --- | --- |
| | Values explored *via* grid search | Best values | Acc. | F1 | ROC-AUC | Precision | Recall |
| Decision Tree | Criterion (gini/entropy), | entropy | 0.885 | 0.903 | 0.890 | 0.938 | 0.871 |
| | max_depth (1, 2,…, 10), | 5 | | | | | |
| | min samples leaf (2, 3, …, 20), | 14 | | | | | |
| | max leaf nodes (1, 2, …, 20) | 17 | | | | | |
| eXtreme Gradient Boosting (XGBoost) for trees | Max depth (5, 6, …, 10), | 5 | 0.892 | 0.909 | 0.893 | 0.933 | 0.887 |
| | alpha (0.1, 0.3, 0.5), | 0.5 | | | | | |
| | learning rate (0.01, 0.02, …, 0.05), | 0.02 | | | | | |
| | estimators (100, 200, 300) | 200 | | | | | |
| Random Forests | Max depth (5, 6, …, 10), | 8 | 0.898 | 0.915 | 0.899 | 0.934 | 0.897 |
| | criterion (gini/entropy), | entropy | | | | | |
| | estimators (100, 200, 300) | 200 | | | | | |

In addition to sources, we examined themes. In line with the typical computational approach used in similar studies on Twitter (*Jelodar et al., 2019*; *Ostrowski, 2015*; *Negara, Triadi & Andryani, 2019*), we use Latent Dirichlet allocation (LDA) to extract topics. To allow for fine-grained comparison, we perform the extraction on every day of the data collection and separate themes of human posts from bots.

# RESULTS

The size of the dataset after preprocessing is provided in the first subsection. For transparency and replicability of research, tweets are available at https://osf.io/3nsf8/; they are organized by key actors and labeled with the sentiment expressed or whether they originate from a bot.

## Preprocessing

In Twitter research, pre-processing often leads to removing most of the data. For example, our previous research on Twitter regarding the Supreme Court (*Sandhu et al., 2019*) discarded 87–89% of the data, while our examination of Twitter and obesity discarded 73% of the data (*Sandhu, Giabbanelli & Mago, 2019*). The reason is that pre-processing has historically involved a series of filters (*e.g.*, removing words that are not deemed informative in English, removing hashtags and emojis), which were necessary as the analysis model could not satisfactorily cope with raw data. In contrast, BERT can directly take tweets as input. Our pre-processing thus only eliminated non-English tweets and tweets with no substantive information that consisted of only hyperlinks. As a result, most of the data is preserved for analysis: we kept 11,007,403 of the original 13,568,750 tweets (Table 7).

**Table 7 Distribution of tweets.**

| Date | Tweets collected | Proportion remaining (%) |
|---|---|---|
| 6 October 2019 | 1,877,693 | 95.3 |
| 17 October 2019 | 1,830,552 | 93.4 |
| 14 November 2019 | 1,828,402 | 92.5 |
| 5 December 2019 | 1,855,771 | 94.6 |
| 19 December 2019 | 1,998,802 | 97.9 |
| 21 January 2020 | 3,230,843 | 96.1 |

| Actor | Name | Post type | | | Sentiment | | |
|---|---|---|---|---|---|---|---|
| | | Bot | Human | Unknown | Negative | Neutral | Positive |
| Adam | Any | 24.10 | 40.90 | 35.00 | 74.10 | 16.70 | 9.20 |
| Schiff | Nickname | 26.80 | 36.10 | 37.10 | 96.80 | 0.20 | 3.10 |
| | Normal | 24.10 | 41.00 | 34.90 | 73.50 | 17.10 | 9.30 |
| Joe Biden | Any | 22.70 | 41.00 | 36.30 | 76.30 | 21.50 | 2.20 |
| | Nickname | 28.10 | 34.50 | 37.30 | 97.80 | 1.90 | 0.40 |
| | Normal | 22.40 | 41.40 | 36.30 | 75.10 | 22.60 | 2.30 |
| Mitch | Any | 22.80 | 49.60 | 27.60 | 64.20 | 33.00 | 2.80 |
| McConnell | Nickname | 16.60 | 40.90 | 42.50 | 94.40 | 4.70 | 1.00 |
| | Normal | 22.80 | 49.60 | 27.60 | 64.10 | 33.10 | 2.80 |
| Nancy | Any | 24.20 | 41.20 | 34.70 | 67.00 | 22.60 | 6.50 |
| Pelosi | Nickname | 26.80 | 36.80 | 36.50 | 99.40 | 0.50 | 0.10 |
| | Normal | 24.10 | 41.30 | 34.60 | 65.90 | 27.40 | 6.70 |

16.60 — 49.60    0.10 — 99.40

**Figure 1 Classification results (sentiments and bots) for each actor.** The high-resolution figure can be zoomed in, using the digital version of this article.

## Sentiment analysis

Analyses shows significant negative sentiment for all key actors (Fig. 1). As expected, when referred to by their nicknames, tweets about the actors were almost unanimously negative. Of the four key actors, Adam Schiff has the most polarizing results, with the highest proportion of positive tweets (9.20%) and the second highest proportion of negative tweets (74.10%). This matches observations during the trial, suggesting that Republicans strongly disliked Schiff for his leading role in the Impeachment Trial, while Democrats supported his efforts. Although the proportions of sentiments expressed vary over time (Fig. 2), we note that negative sentiments dominate on every one of the six dates for data sampling.

## Bot detection

Results from bot detection show that at least 22–24% of tweets were sent by bot controlled accounts, with a few examples provided in Table 8. Deleted accounts as well as accounts for which no information was obtained are grouped together in the 'Unknown' category. As Twitter will remove bots that manipulate conversations, it is possible that a significant share of the 'Unknown' category is also composed of bots. We note a

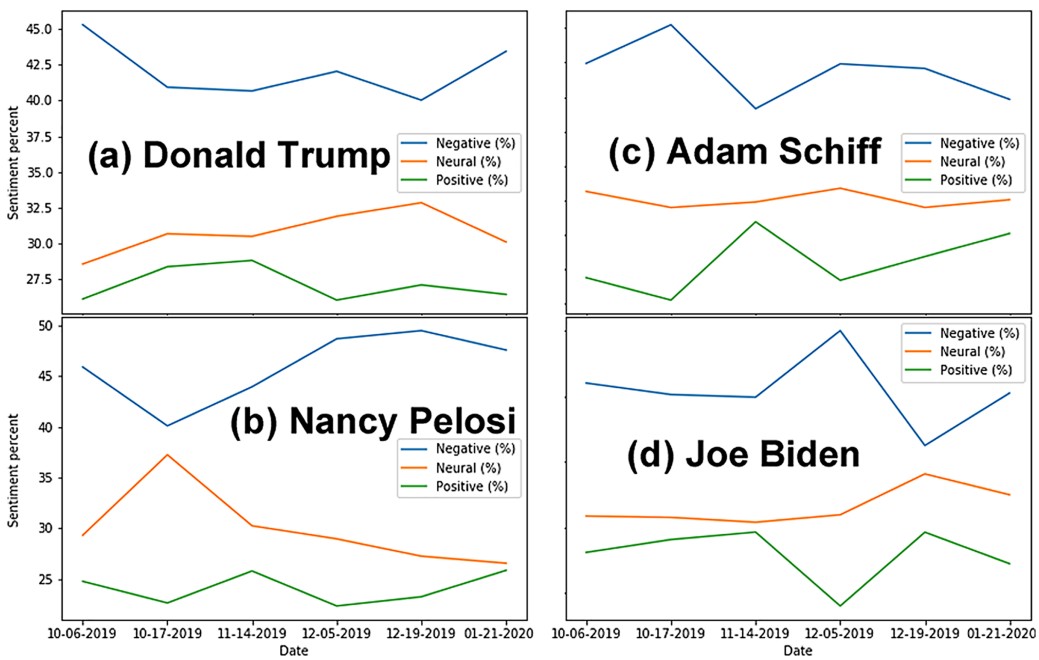

**Figure 2** **Percentage of tweets per category of sentiments over time for each actor.** (A) Donald Trump. (B) Nancy Pelosi. (C) Adam Schiff. (D) Joe Biden. Note that using a *percentage* instead of the absolute number of tweets allows to compare results across actors, since each one is associated with a different volume of tweets.

difference between results for the Republican Mitch McConnell, on the one hand, and the three Democrats (Joe Biden, Nancy Pelosi, and Adam Schiff) on the other hand. For Democrats, bots created at least 26–28% of the more derogatory tweets (*i.e.*, based on nicknames), whereas this percentage falls to 16% when referring to Mitch McConnell by his nickname. Overall, a greater proportion of tweets on Mitch McConnell (by almost 10 percentage points) were made by humans, compared to Democrats. Further analyses on other political figures confirmed that bots targeted Democrats (Jerry Nadler 25.83%, Chuck Schumer 24.40%, Hakeem Jeffries 23.55%) more often than Republicans (Lindsey Graham 22.08%, Pat Cipollone 22.46%, Josh Hawley 15.93%, Stephen Miller 22.14%), with an average difference of four percentage points.

Although the overall tone of the tweets is negative (Fig. 3), the *ratio of negatives-to-positives* shows a clear differentiation in the use of bots compared to humans. For example, consider Adam Schiff (Fig. 3, top-left). Based on tweets by humans, he has a ratio of $\frac{583616}{70666} \approx 8.3$ negatives-to-positives. In comparison, this ratio is $\frac{354241}{36542} \approx 9.7$ for tweets authored by bots. Overall, we observe that the only cases in which bots had a greater negative-to-positive ratio of posts than humans was for Democrats: Adam Schiff (9.7 negative tweets by bot for each positive *vs.* 8.3 for humans), Chuck Schumer (40.2 *vs.* 33.4), Joe Biden (39.1 *vs.* 33.8), and Nancy Pelosi (10.9 *vs.* 10.2). In contrast, bots were always less negative than humans for Republicans: Donald Trump (6.1 for bots *vs.* 7.9 for humans), Lindsey Graham (8.1 *vs.* 9.0), Mitch McConnell (19.7 *vs.* 23.6), and Stephen Miller (59.6 *vs.* 103.8).

**Table 8 Two sample tweets for each day of data collection.**

| Date | Tweets |
|---|---|
| 5 December 2019 | RT @usa4_trump: Ukraine Pulls Back Curtain On Biden – Claims Burisma Paid The Vice Presi… https://t.co/lhGlqOaNhn *via* @YouTube HERE IT CO… |
| | @GOP @RepMikeJohnson Too much evidence to ignore. Trump won't even let staff testify. No transparency, no innocence! |
| 19 December 2019 | Merry Christmas Eve Patriots! I have a grateful heart for President Donald Trump, a man who loves our God and our country, and defends our people. Keep the faith Patriots, and know that we are on the right path. God Bless you all. #TRUMP2020Landside https://t.co/y5duaB6wrR |
| | @RyanHillMI @realDonaldTrump You're right. It wasn't a trial. It was a perfect example of tyranny. But, tweets can only be so long and impeachment didn't fit. So, trial. |
| 6 October 2019 | @AndyOstroy @realDonaldTrump What Law has been broken? What rule? Why are they not taking a formal vote to begin impeachment? What crime or misdemeanors took place? Pelosi cant articulate the actual crime! Schiff is 4–50 pinnochios deep in 3 years! |
| | Jim Jordan Doesn't 'Think' Trump Did Something He Actually Did On Camera https://t.co/TjXU4chJ5T |
| 17 October 2019 | Survey: 54 percent of Americans support Trump impeachment inquiry https://t.co/0eYzwSn03l: Hillary beat Trump by a majority of votes. Polls show, that disgruntled Hillary voters, want the W.H. back, there will always be a majority against Trump ??? |
| | RT @weavejenn: Cause he's a spineless coward #TrumpIsADisgrace https://t.co/6RhP0UfId9 |
| 14 November 2019 | @CumeTalitha trump is our god and lord and savior. dont say gods name in vein. i will report you. |
| | @realDonaldTrump Adam Schiff is the fake whistle blower! No one exists all made up! And what are they whistle blowing we have the transcripts end of story. |
| 21 January 2020 | @CHHR01 Pence also tried to oust Trump from the ticket in 2016. They even created a website, but then abandoned the effort and threatened the life of the web designer. I will find that story for you. |
| | @realDonaldTrump FATHER TODAY I ASK FOR A MIRACLE FOR PRWSIDENT TRUMP LET HIM SEE ALL HIS ENEMYS HANDCUFFED ESCORTED TO GITMO4 TREASON IN JESUSNAME.REMOVE SOROS ALL THE DEMOCRATES WHO WE SEE DESTROYING CALI NEW YORK CHICAGO ALL STATES/CITYS(ISA41:11-16)JNA |

**Note:**
Some links may have broken since the study was conducted, which is one indicator that they were propaganda posts.

We did not find evidence of a clear temporal trend (*e.g.*, monotonic increase or decrease) of bots over time (Fig. 4), which may suggest that they were employed based on specific events.

## Top sources used by bots

A total of 1,171 distinct websites were used as sources across 54,899 tweets classified as written by bot accounts. The complete list is provided as part of our supplementary online materials (https://osf.io/3nsf8/). The distribution of sources is heavily imbalanced. The vast majority ($n = 909$, 77.62%) of these websites were used less than 10 times each, while the top 10 websites were used 32,050 times (*i.e.*, in 58.38% of tweets with URLs). The most cited website is Twitter (19,258 tweets, 35.08%), as expected by the practice of retweets. Several other websites are social media (YouTube in 1,884 tweets or 3.43%, Facebook in 266 tweets or 0.48%, Periscope in 215 tweets or 0.39%), web hosting solutions (WordPress in 111 tweets or 0.20%, Change.org in 106 tweets or 0.19%, Blogspot in 101 tweets or 0.18%), or search engines (Civiqs in 65 tweets or 0.12%, Google in 65 tweets or 0.12%). The remainder are primarily websites that propose news, hence they were evaluated with respect to the bias ratings from AllSides and Ad Fontes.

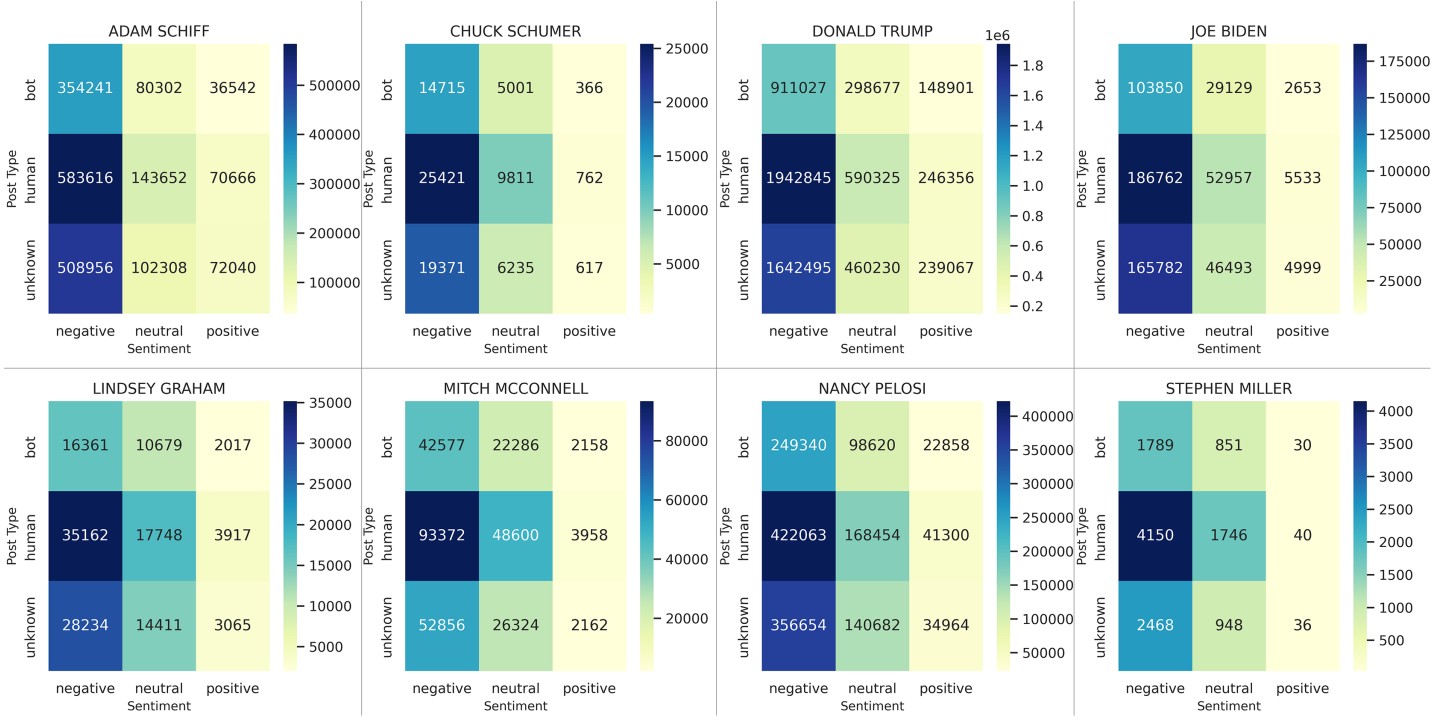

**Figure 3 Joint examination of sentiments and bots for each major actor, with additional political figures included.** The high-resolution figure can be zoomed in, using the digital version of this article.

Given the large number of distinct websites, we focused on those used at least 50 times, excluding the categories aforementioned (social media, web hosting, search engines). When using AllSides, the sample *appears* relatively balanced (Table 9), because a large number of websites are not rated. Ad Fontes provides more evaluations, which starts to show that websites lean to the right. We complemented the ratings of Ad Fontes with manual evaluations (Table 9), showing that (i) there are almost twice as many right-leaning sources than left-leaning sources and (ii) the right-leaning sources are much more extreme. For example, 41 websites are more right-leaning than FoxNews while only 16 were more left-leaning than CNN. The imbalance is even more marked when we consider the volume of tweets. Indeed, out of the 10 most used websites, eight lean to the right and two to the left (italicized): FoxNews (3.73% of tweets with sources), Trending Politics (2.80%), Breitbart (2.60%), The Right Scoop (2.49%), The Gateway Pundit (2.48%), Sara A Carter (2.19%), Wayne Dupree (1.86%), *CNN* (1.69%), *Raw Story* (1.57%), Kerry Picket (1.41%). Tweets associated with some of these websites are provided in Table 10 as examples. In sum, there is ample evidence that the bots tend to promote a Republican viewpoint, often based on misleading sources.

Our findings are similar to *Tripodi & Ma (2022)*, who noted that almost two thirds of the sources in official communication from the White House under the Trump Administration relied on "RWME, a network fueled by conspiracy theories and fringe personalities who reject normative journalistic practices." Although other news outlets

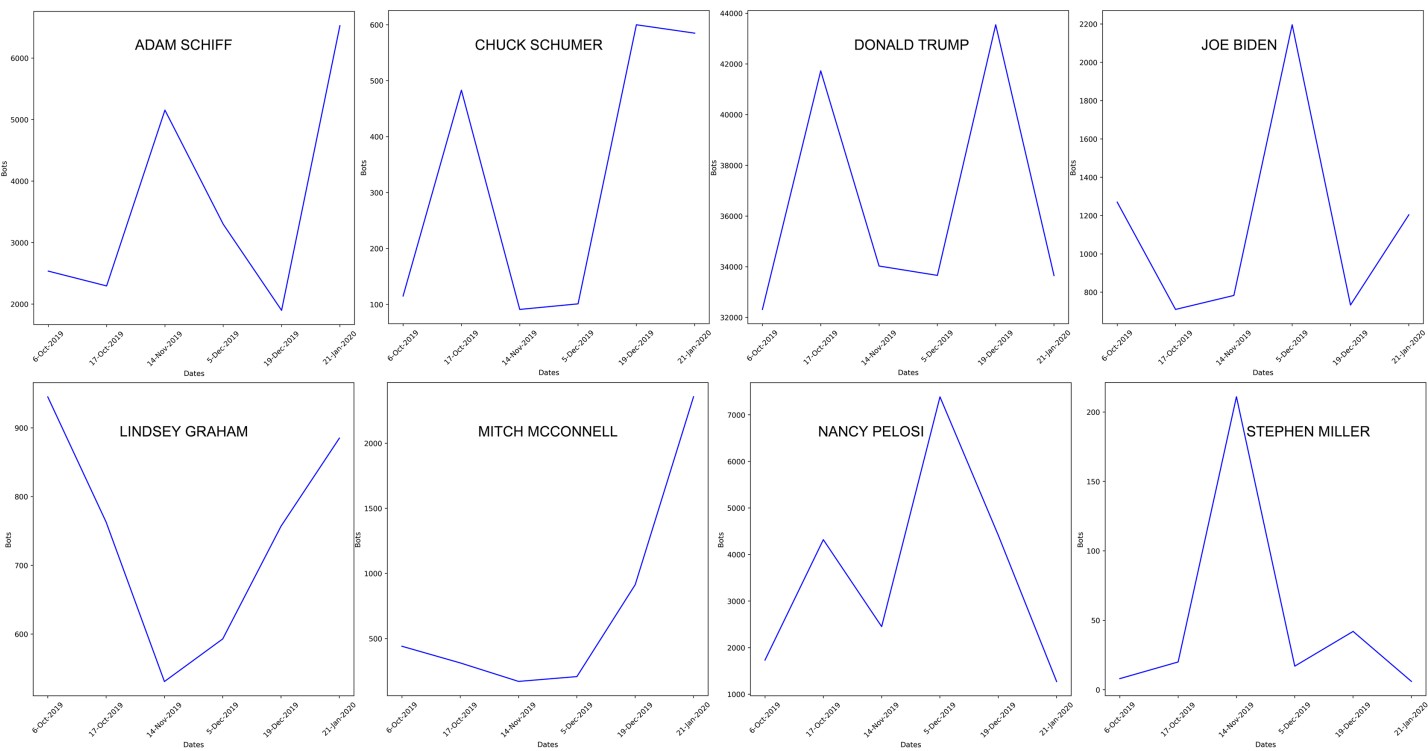

**Figure 4 Temporal trend in the number of posts by bots for each major actor, with additional political figures included.** The high-resolution figure can be zoomed in, using the digital version of this article.

**Table 9 Website categories per source for bias rating (AllSides, Ad Fontes), using their nomenclature (*e.g.*, 'very left' in AllSides, 'hyper-partisan left' in Ad Fontes).** *Manual screening was applied to websites not rated by Ad Fontes.

| Leaning | AllSides | Ad Fontes | Manual* | Ad Fontes + Manual | *Overall* |
|---|---|---|---|---|---|
| Not rated | 40 | 16 | | | |
| Very left/hyper-partisan left | 9 | 6 | | 6 | Left: 25 |
| Left | 12 | 19 | | 19 | |
| Center/balanced | 8 | 13 | 1 | 14 | |
| Right | 5 | 16 | 1 | 17 | Right: 46 |
| Very right/hyper-partisan right | 14 | 16 | | 16 | |
| Most extreme right | | 2 | 11 | 13 | |
| No longer accessible | | | 3 | 3 | |

were cited, such as the Washington Post and CNN, this was often in the context of characterizing them as 'fake news'. Other recent analyses have also noted that the use of popular far-right websites was more prevalent in counties that voted for Trump in 2020 (*Chen et al., 2022*). The possibility for websites that fuel partisanship and anger to be highly profitable may have contributed to their proliferation (*Vorhaben, 2022*).

**Table 10 Sample tweets pointing to hyper-partisan right or extreme right websites.**

| Website | Tweet |
| --- | --- |
| rightwingtribune.com | Hes one to ca'll the kettle black, Where's the whistle blower there Schifty?? You said there was, then he wasn't, you werent clear either, but Pelosi had your backside, didn't she? You, all the other DemocRATS. @realDonaldTrump @GodLovesUSA1 @FogCityMidge https://t.co/6jihkrJrXM |
| twitchy.com | Straightforward from here! Vox sounds pretty convinced that impeachment could ultimately lead to President Nancy Pelosi https://t.co/qFUY9aPyKX *via* @twitchyteam |
| thegatewaypundit.com | RT @gatewaypundit: REMINDER: PELOSI AND SCHIFF BOTH CONNECTED TO UKRAINIAN ARMS DEALER @JoeHoft https://t.co/bTIxwGZ4gy *via* @gatewaypundit |
| therightscoop.com | RT @marklevinshow: Fascistic Nancy Pelosi, also the dumbest speaker https://t.co/PbCWnWGeHs |
| saraacarter.com | RT @SaraCarterDC: Trump Calls Speaker Pelosi 'A Third-Rate Politician' https://t.co/9jd3MzOuH9 |
| bongino.com | This Says it All: Watch Chairman Nadler Fall ASLEEP During Impeachment Hearing | Dan Bongino https://t.co/LySASgD8vF |

**Note:**
Some links may have broken since the study was conducted, which is one indicator that they were propaganda posts.

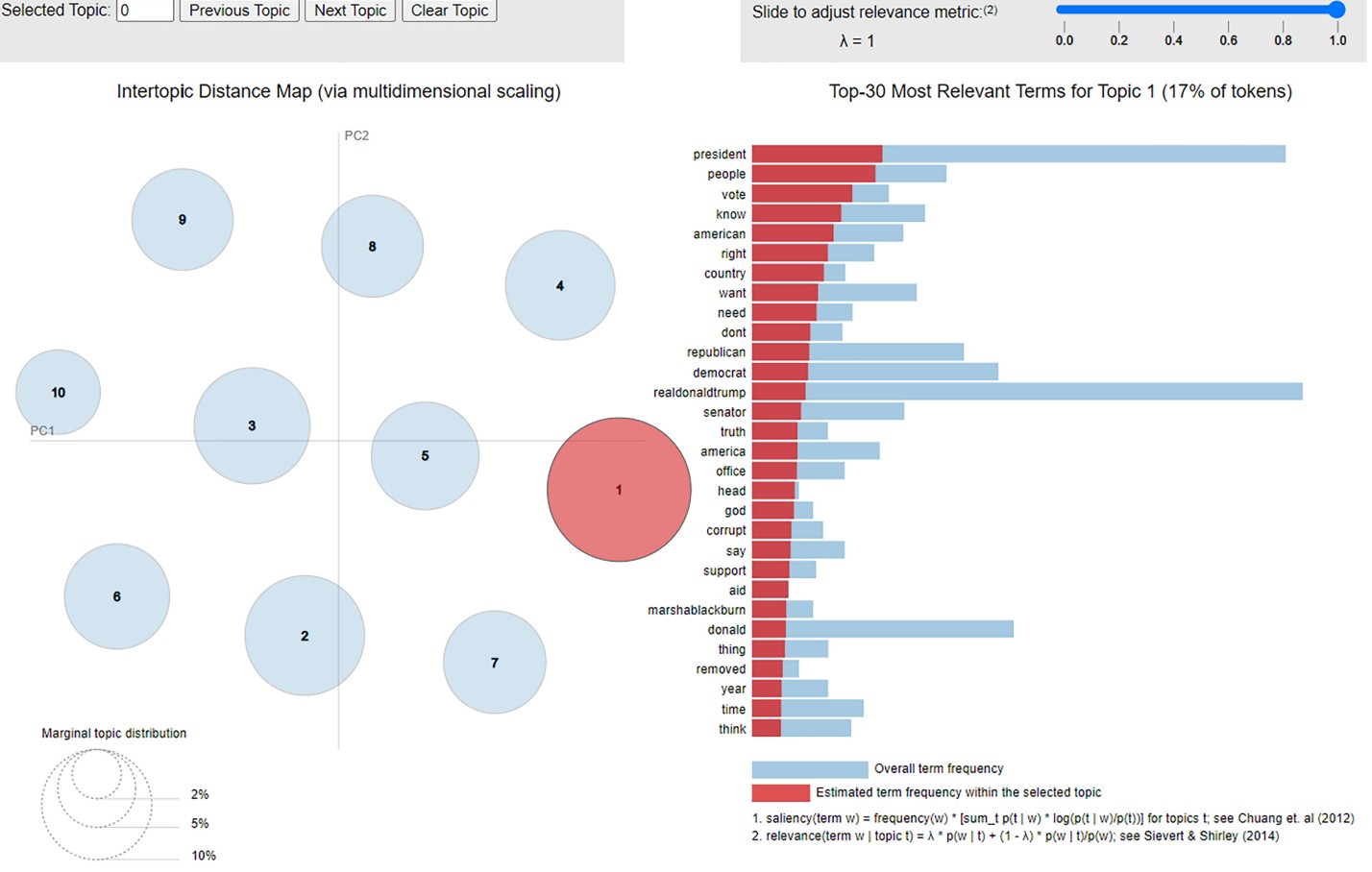

**Figure 5 Interactive visualization for topics from bots on 21 January 2020.**

## Themes for humans *vs.* bots

Interactive visualizations for themes are provided as part of our supplementary online materials (https://osf.io/3nsf8/), with one example shown in Fig. 5. Themes should be

interpreted *with caution*. For example, consider posts on 6 December 2019. For bots, a major theme links Senator Nancy Pelosi with 'hate', whereas for humans she is first associated with impeachment, house, president, and then only with hate. However, bots are not actually conveying images of hatred *towards* Pelosi. Rather, the connection is related to a news story that occurred the day before, in which a reporter asked Pelosi if she hated Trump, and she responded that she did not hate anyone; the reaction went viral. Although bots have picked up on it more than the humans, it is only tangentially related to impeachment and does not indicate that bots are hating Pelosi.

A more robust and relevant pattern, across the entire time period, is that Biden is a much more salient term for the humans than the bots. Based on associated terms, it appears to be connected to Trump and other Republicans (particularly Lindsey Graham and outlets such as Fox News) trying to distract from impeachment by making allegations that Biden was corrupt in his dealings with Ukraine. It thus appears that bots and humans occasionally focus on different stories.

## DISCUSSION

Throughout the process, both parties and the President tried to control the public narrative over impeachment. Early in the process, Democrats began to talk about abuse of power or corruption by Trump and eventually shifted to the more specific language of bribery and extortion. Meanwhile, Republican elected officials seemed to be trying to frame this as a coup by the Democrats. Republicans directed their ire at House Intelligence Committee Chairman Democrat Adam Schiff who led the impeachment inquiry. As summarized by Barberio in the context of power scandals, "Trump's approach to scandal management included the contemplation of the use of backfires and their full deployment" (*Barberio, 2020*). The President and some of his Republican allies thus tried to divert attention from the President by alleging misconduct by Joe and Hunter Biden. With the 2020 Democratic presidential primary campaigns well underway, Trump saw an opportunity to undermine Biden's presidential campaign by raising doubts about Biden's integrity. In early October, the Trump campaign demonstrated a willingness to influence perceptions *via* social media, by spending "as much as $21,000" on a Facebook ad entitled "Biden Corruption", which included debunked claims that then Vice President Joe Biden had tied Ukrainian aid to dropping an investigation of his son (*Grynbaum & Hsu, 2019*). Throughout the process, the President denigrated Democrats as a group and individually in his public comments and on Twitter. Strategies have included the use of nicknames, which contributes to painting a gallery of 'known villains' (*Montgomery, 2020*) and fits within the broader pattern of the far right in online harassment of political enemies (*Bambenek et al., 2022*). This use of nicknames also echoes Trump's strategy in employing derogatory language when depicting other groups, such as the frequent use of terms such as 'animal' or 'killer' when referring to immigrants (*Bilewicz & Soral, 2022*). Such strategies are part of a broader trend of incivility by American politicians on Twitter, with uncivil tweets serving as powerful means of political mobilization and fundraising (*Ballard et al., 2022*) by drawing on emotions such as anger (*Joosse & Zelinsky, 2022*; *Bernecker, Wenzler & Sassenberg, 2019*); the ensuing attention may even lead politicians to engage

into greater incivility (*Frimer et al., 2022*). In the case of Trump, the violence-inducing rhetoric resulted in permanent suspension of his Twitter account, days before being charged by the House of Representatives for "incitement of insurrection" (*Wheeler & Muwanguzi, 2022*).

Previous research on Trump and social media mining concluded that "political troll groups recently gained spotlight because they were considered central in helping Donald Trump win the 2016 US presidential election" (*Flores-Saviaga, Keegan & Savage, 2018*). Far from isolated individuals with their own motives, research has shown that 'trolls' could be involved in coordinated campaigns to manipulate public opinion on the Web (*Zannettou et al., 2019*). Given the polarization of public opinions and the "hyper-fragmentation of the mediascape", such campaigns frequently tailor their messages to appeal to very specific segments of the electorate (*Raynauld & Turcotte, 2022*). A growing literature also highlights that the groups involved in these campaigns are occasionally linked to state-funded media (*Sloss, 2022*), as seen by the strategic use of Western platforms by Chinese state media to advance 'alternative news' (*Liang, 2021*) with a marked preference for positives about China (*Wu, 2022*) or the disproportionate presence of bots among followers of Russia Today (now RT) (*Crilley et al., 2022*).

In this paper, we pursued three questions about bots: their overall involvement (Q1), their targets (Q2), and the sources used (Q3). We found that bots were actively used to influence public perceptions on online social media (Q1). Posts created by bots were almost exclusively negative and they targeted Democrat figures more than Republicans (Q2), as evidenced both by a difference in ratio (bots were more negative on Democrats than Republicans) and in composition (use of nicknames). While the effect of a presidential tweet has been debated (*Miles & Haider-Markel, 2020*), there is evidence that such an aggressive use of Twitter bots enhances political polarization (*Gorodnichenko, Pham & Talavera, 2021*). Consequently, the online political debate was not so much framed as an exchange of 'arguments' or support of various actors, but rather as a torrent of negative posts, heightened by the presence of bots. The sources used by bots confirm that we are not in the presence of arguments supported by factual references; rather, we witnessed a large use of heavily biased websites, disproportionately promoting the views of partisan or extreme right groups (Q3). Since social media platforms are key drivers of traffic towards far-right websites (together with search engines) (*Chen et al., 2022*), it is important to note that many of the links towards such websites are made available to the public through bots. Although our findings may point to the automatic removal of bots from political debates as a possible intervention (*Cantini et al., 2022*), there is evidence that attempts at moderating content can lead to even greater polarization (*Trujillo & Cresci, 2022*), hence the need for caution when intervening.

While previous studies found a mostly anti-Trump sentiment in human responses to tweets (*Roca-Cuberes & Young, 2020*), the picture can become different when we account for the sheer volume of bots and the marked preference in their targets. This use of bots was not an isolated incident, as the higher use of bots favoring Trump compared to

Clinton was already noted during the second U.S. Presidential Debate (*Howard, Kollanyi & Woolley, 2016*), with estimates as high as three pro-Trump bots for every pro-Clinton bot (*Ferrara, 2020*). Our analysis thus provides further evidence for the presence of 'computational propaganda' in the US.

There are three limitations to this work. First, it is possible that tweets *support* an individual despite being *negative*. However, this effect would be more likely for Republicans than Democrats, which does not alter our conclusions about the targeted use of bots. Indeed, Trump's rhetoric has been found to be "unprecedentedly divisive and uncivil" (*Brewer & Egan, 2021*), which can be characterized by a definite and negative tone (*Savoy, 2021*). Consequently, some of the posts categorized as negatives may not be *against* Trump, but rather amplify his message and rhetoric to *support him*. The prevalence of such cases may be limited, as a recent analysis (over the last two US presidential elections) showed that the prevalence of negative posts was associated with lower popular support (*Shah, Li & Hadaegh, 2021*).

Second, although we observe an overwhelmingly negative tone in posts from bots, there is still a *mixture* of positive and negative tweets. This mixture may reflect that the use of bots in the political sphere is partly a state-sponsored activity. That is, various states intervene and their different preferences produce a heterogeneous set of tweets (*Kießling et al., 2019*; *Stukal et al., 2019*). Although studies on centrally coordinated disinformation campaigns often focus on how one specific entity (attempts to) shape the public opinion (*Keller et al., 2020*; *Im et al., 2020*), complex events involve multiple entities promoting different messages. For example, analysis of the 2016 election showed that IRA trolls generated messages in favor of Trump, whereas Iranian trolls were against Trump (*Zannettou et al., 2019*). It is not currently possible to know with certainty which organizations were orchestrating which bot accounts, hence our study cannot disentangle bot posts to ascribe their mixed messages to different organizations.

Finally, in line with previous large-scale studies in natural language processing (NLP) regarding Trump, we *automatically* categorized each post with respect to sentiment (*Tachaiya et al., 2021*). Although this approach may involve manual reading for a small fraction of tweets (to create an annotated dataset that then trains a machine learning model), most of the *corpus* is then processed automatically. Such an automatic categorization allows analysts to cope with a vast amount of data, thus favoring breadth in the pursuit of processing millions of tweets. In contrast, studies favoring depth have relied on a qualitative approach underpinned by a *manual* analysis of the material, which allows examination of the 'storytelling' aspect through changes in rhetoric over time (*Phelan, 2021*). As our study is rooted in NLP and performed over millions of tweets, we did not manually read them to construct and follow narratives. This may be of interest for future studies, who could use a complementary qualitative approach to contrast the rhetoric of posts from bots and humans, or how bots may shape the nature of the arguments rather than only their polarity. Such mixed methods studies can combine statistical context analysis and qualitative textual analysis (*O'Boyle & Haq, 2022*) to offer valuable insights.
## CONCLUSIONS

We found that bots have played a significant role in contributing to the overall negative tone of the debate during the first impeachment. Most interestingly, we presented evidence that bots were targeting Democrats more than Republicans. The sources promoted by bots were twice as likely to espouse Republican views than Democrats, with a noticeable presence of highly partisan or even extreme right websites. Together, these findings suggest an intentional use of bots as part of a larger strategy rooted in computational propaganda.

## ACKNOWLEDGEMENTS

We thank Z. Li, B. Lee, J. Holt, and D. Smith, for their assistance with this project. We also thank Dr. J. Mueller for supporting our use of the Redhawk Computing Cluster.

### Funding

The authors received no funding for this work.

### Competing Interests

The authors declare that they have no competing interests.

### Author Contributions

- Michael C. Galgoczy performed the experiments, analyzed the data, performed the computation work, prepared figures and/or tables, and approved the final draft.
- Atharva Phatak performed the experiments, analyzed the data, performed the computation work, prepared figures and/or tables, and approved the final draft.
- Danielle Vinson conceived and designed the experiments, authored or reviewed drafts of the paper, and approved the final draft.
- Vijay K. Mago conceived and designed the experiments, authored or reviewed drafts of the paper, and approved the final draft.
- Philippe J. Giabbanelli conceived and designed the experiments, analyzed the data, prepared figures and/or tables, authored or reviewed drafts of the paper, and approved the final draft.

### Data Availability

The code and data are available at the Open Science Framework (OSF):

Giabbanelli, Philippe J. 2022. "Trump Twitter." OSF. February 17. osf.io/3nsf8.

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
