# Peer review of "(Re)shaping online narratives: when bots promote the message of President Trump during his first impeachment"

_PeerJ Computer Science, doi:10.7717/peerj-cs.947_

## Round 0.1 · original submission · Major Revisions

The manuscript requires improvement in the explanation of the presented method.

·

Basic reporting

Hi,
I hope you all are doing well.
The work by the authors is good and the presentation of data is also good, personally, I like the topic.
The author should improve the article to make it significant as I am suggesting some points.

1- Abstract is too long and even the main points of studies are missing in the abstract. it's too general.
2- Add contributions in the introduction section with bullets.
3- Literature is missing at least add some NLP-related studies. I am suggesting some below:
a- A performance comparison of supervised machine learning models for Covid-19 tweets sentiment analysis
b- Tweets classification on the base of sentiments for US airline companies
c- Determining the Efficiency of Drugs under Special Conditions from Users’ Reviews on Healthcare Web Forums
d- US Based COVID-19 Tweets Sentiment Analysis Using TextBlob and Supervised Machine Learning Algorithms
e- Sentiment Analysis and Topic Modeling on Tweets about Online Education during COVID-19
f- Deepfake tweets classification using stacked Bi-LSTM and words embedding

4- Add some description of the used model's architectures.
5- Add comparison with Textblob and perform the comparison of machine learning models with textblob sentiments.
6- We didn't used neural networks models?

Experimental design

Mention above

Validity of the findings

Mention above

Additional comments

Mention above

Reviewer 2 ·

Basic reporting

The article provides a novel approach to the treatment of bots, taking the learnign machine technique as the main quantitative measurement tool.
However, the authors should consider the following requirements and recommendations in order to improve all areas.

Title: I woud add the date of the political trial to the title to diferentiate from other impeachments.

Abstract: The objective of the research should be clearer in the abstract, in line with the established hypothesis.

The study should be expanded in future research and the current limitations should be overcome.

This type of study, more than the numerical value, requires the qualitative and discursive-rhetorical analysis of the content of the bots.

It does not matter if it is necessary to reduce the sample and resort to manual measurement.

The final selection of actors is not clear; in the Discussion section in Figure 3 eight actors are discussed ... In previous graphs, only four are discussed.
They should also be reflected in the abstract and in the Method (selected sample).

Is the sample considered balanced based on the ideology and membership of the Democratic and Republican parties of these actors?

Literature: The literatura missing references to the use of disinformation, political corrupción, twitter-rhetoric and critical analysis of speech.

In general, we value the updated bibliography that is provided related to the study.
Some recommended papers are related in the bibliography.

As the article focuses on the Twitter network, we consider the following quote relevant:
Campos-Domínguez, Eva (2017). “Twitter y la comunicación política”. El profesional de la información, v. 26, n. 5, p p . 785-793.
https://doi.org/10.3145/epi.2017.sep.01

Experimental design

Research questions are not defined, essential in the methodological section of a study with these characteristics.

The criteria for selecting the sample of political actors analyzed are not clearly described.

A variable sheet is recommended to help identify which measurement elements have been taken into account:
Actors
Post Type: Bot, Human, Unknown
Type of feeling: positive, negative, neutral ...


Method:

I would recommend to apply the triangulated content analysis methodology with a triple approach (qualitative, quantitative, discursive) on a smaller sample of bots.

The fact of explaining the variables in the methodology would help to better understand the subsequent tables and graphs and the results of the analysis.

We value the justified selection of sampling dates.
As keywords, It would include the word disinformation, fallacy, lie, or something that reflects the purpose of the bot.


Define the selection criteria for these actors; see if there is a balance in the selection, between Democratic and Republican leaders.

The arbitrary selection of 3000 tweets enables a study of qualitative and discursive marks, which would get better the methodology of the article.

We recommend incluiding:

-Krippendorff, Klaus (2004). Content analysis. Sage. ASIN: B01B8SR47Y

-Van-Dijk, Teun A. (2015). “Critical discourse studies. A sociocognitive Approach”, Methods of Critical Discourse Studies, v. 3, n. 1.

-Pérez‐Curiel, Concha, Rivas‐de‐Roca, Rubén; García‐Gordillo, Mar (2021). “Impact of Trump’s Digital Rhetoric on the US Elections: A View from Worldwide Far‐Right Populism”, Social Sciences, v. 10, n. 152.
https://doi.org/10.3390/ socsci100501

We value the detailed technical explanation of using botometer.

Validity of the findings

Working with most of the data is appreciated; differential value of BERT.

However, the results of a bot analysis should not be limited only to the quantitative, but should explain the topic of the bot, the profile of actors it praises or attacks ... if the bot includes hyperlinks, know what content they do reference...
If the study also proposes an analysis of feelings, it is considered advisable to consult other investigations focused on the use of the fallacy and the propaganda of political discourse:

Pérez-Curiel, Concha; Velasco-Molpeceres, Ana-María (2020). "Impact of political discourse on the dissemination of hoaxes about Covid-19. Influence of misinformation in the public and the media ”, Revista Latina de Comunicación Social, n. 78, pp. 65-97.
https://www.doi.org/10.4185/RLCS-2020-1469


More than a recommendation, it could be considered as a need to complete the quantitative part, linked exclusively to machine learning.

It would be appropriate to include some captures of Twitter messages cataloged as Bots.

Results:

The Figure 1 is very interesting. I would delve further into the interpretation of it.

For example, analyzing the authorship factor of the post. The value of Human ahead of Bot or Unknowwn.

It would also be convenient to interpret the tone of the sentiment shown in each message beyond the numerical.

The question is: what variables determines the feeling, considering that it is a subjective factor?

Very interesting the study of the tone of feelings.

Discussion:

Correct use of antecedents and previous context to justify the origin of the bots

It is necessary to explain the criterion of why in some tables the study is limited to four actors and in others (Figure 3) a larger sample is taken.

The limitations of the type of method used are obvious; the study is complete but only if we consider quantitative measurement. When the object of study are bots or false messages spread with the strategy of influencing, discursive analysis is required as the main methodology. It is highly recommended to keep this perspective in mind for present and future research.

Conclusions:

The conclusions should be completed much more in spite of the fact that the discussion is quite developed.

One possibility is to link the conclusions to the discussion, at the same point.

It is important that the conclusions refer to research questions that are not detailed in this study (see comment on methodology).

They could also turn these conclusions around the objectives of the investigation. Neither are they well defined in the abstract or the text of the article.
Considering that only quantitative measurement is performed, the significant role of bots in intensifying negative tone during impeachment cannot be affirmed. Numerical values ​​cannot measure message intentionality.
The conclusions also refer to the most relevant literature on which the study has been based and to what extent this research provides novelty compared to previous ones.

The conclusions should be completed much more in spite of the fact that the discussion is quite developed.

One possibility is to link the conclusions to the discussion, at the same point.

It is important that the conclusions refer to research questions that are not detailed in this study (see comment on methodology).

They could also turn these conclusions around the objectives of the investigation. Neither are they well defined in the abstract or the text of the article.
Considering that only quantitative measurement is performed, the significant role of bots in intensifying negative tone during impeachment cannot be affirmed. Numerical values ​​cannot measure message intentionality.
The conclusions also refer to the most relevant literature on which the study has been based and to what extent this research provides novelty compared to previous ones.

Additional comments

The article should generally deal more in depth with the literature related to the proposed topic.
In addition, it is essential to explain the variables, the sample and ask whether the sentiment analysis requires a qualitative and descriptive approach, as well as a qualitative one.

Annotated reviews are not available for download in order to protect the identity of reviewers who chose to remain anonymous.

Reviewer 3 ·

Basic reporting

In this manuscript, the authors examine the role of bots in the spreading of Donald Trump’s tweets during the first impeachment trial. Specifically, the authors collected and analyzed more than 13 million tweets, collected during six key dates during the impeachment trial, to perform (1) sentiment analysis and (2) bot detection. Overall, the setup of the study and the analysis of the data were done well. I recommend Accept.

Experimental design

To assess activities on Twitter during Trump’s first impeachment trial, the authors collected more than 13 million tweets on six key dates. For the sentiment analysis portion, the authors used a supervised learning approach, via the natural language processing model BERT. Though I was initially concerned with using only a training set of 3,000 tweets and applying the model to such a large dataset, the authors alleviated those concerns with adequate citations and associated discussions. For the bot detection portion, the authors used the botometer library and applied three different supervised learning techniques to the data. Overall, the research design was well-crafted, the methodology detailed and rigorous, and the discussion thorough.

Validity of the findings

Data is provided via online repository. The results presented in the manuscript are sound and follow from the analyses. Statistically, the results and analyses are sound and rigorous. Consistent with their research question, the authors show that bots contributed significantly to the negative rhetoric on Twitter. Not only do bots comprise of approximately 20-25% of all tweets, bots are also overwhelmingly negative and appeared to be strategic in nature. Bots not only targeted Democratic politicians more than Republicans, tweets directed at Democratic politicians were more negative than those targeted at Republican politicians. More broadly, these results add to our understanding of political communications on Twitter, as well as the impact of bots on the political system.

Additional comments

• Misspelling of the word “analyses.” In the manuscript, the authors misspelled the noun “analyses” as “analyzes.” See, for instance, line 15, 66, and 67.
• Unnecessary capitalizations. For example, the word “tweets” is generally lowercase, but is uppercase in multiple spots in this manuscript (line 123, etc.)
• Misspelling of the word “Twitter.” In the manuscript, Twitter is sometimes misspelled as “Tweeter.” (line 139)

---

## Round 0.2 · Minor Revisions

Kindly improve the manuscript as per the reviewers suggestions and comments.

·

Basic reporting

Authors have done good work in revision and my recommendation for the paper is accepted.

Experimental design

No Comments

Validity of the findings

Comments

Additional comments

Comments

Reviewer 2 ·

Basic reporting

The authors have made the requested modifications.
These changes greatly improve the quality of the proposal.

Experimental design

The methodology is now described in a more detailed and rigorous way.
Introducing research questions has helped to better structure the results and the discussion.

Validity of the findings

The results section has improved considerably as well as the discussion.
Following the instructions of the reviewer, the position of the different parties, the type of bias, the authorship of the bots, among other topics of interest, are argued.
Bearing in mind that the data is adequately analyzed, the discussion introduces a justified interpretation and reflection. Therefore, the conclusion is related to the objectives and research questions.

Additional comments

We thank the authors for all the changes made to improve the proposal.

Reviewer 3 ·

Basic reporting

This article is well-written, with some minor errors (detailed later). The authors adequately addressed the suggestions made by other reviewers and myself from the initial round of review.

Experimental design

In my original review, I summarized, “Overall, the research design was well-crafted, the methodology detailed and rigorous, and the discussion thorough.” The revised manuscript, to address the questions raised by other reviewers, is even better.

Validity of the findings

I like the revised manuscript. The authors made clearer their contributions (i.e., the three research questions), and the presentation of the empirical results is well-aligned and presented to the noted contributions.

Additional comments

In this manuscript, the authors examine the role of bots in the spreading of Donald Trump’s tweets during the first impeachment trial. Specifically, the authors collected and analyzed millions of tweets, collected during six key dates during the impeachment trial, to perform (1) sentiment analysis and (2) bot detection. Specifically, the authors asked: (1) Are bots actively involved in the debate? (2) Do bots target one political affiliation more than another? And (3) Which sources are used by bots to support their arguments? In my original review, I recommend Accept. The revised manuscript is significantly better. After reading the comments from other reviewers and the authors’ response, I again recommend acceptance of this manuscript, with suggestions below for minor revisions.

Minor points
• Discrepancy in number of tweets collected. In the abstract, the authors noted that they collected more than 10 million tweets. However, in the Methods section (Line 141), the authors noted they collected more than 13 million tweets. This is likely a typo, but it needs to be corrected
• To be consistent with official spelling, I recommend change these company names: “youtube” to “YouTube,” “facebook” to “Facebook,” “civiqs” to “Civiqs,” “google” to “Google,” “wordpress” to “WordPress,” “change.org” to “Change.org,” “blogspot” to “Blogspot,” etc. This specific comment is particular to Lines 308-315. Interestingly, company names are correctly capitalized in Lines 316-330, so I recommend revising to be consistent. I also recommend checking the rest of the manuscript (text, figures, tables, etc.) for these issues.

---

## Round 0.3 · accepted · Accept

The authors have successfully addressed the reviewer comments.